# Contemporary Approaches to the Surgical Management of Pancreatic Neuroendocrine Tumors

**DOI:** 10.3390/cancers16081501

**Published:** 2024-04-14

**Authors:** Akash Kartik, Valerie L. Armstrong, Chee-Chee Stucky, Nabil Wasif, Zhi Ven Fong

**Affiliations:** Division of Surgical Oncology and Endocrine Surgery, Department of Surgery, Mayo Clinic Arizona, Phoenix, AZ 85054, USA

**Keywords:** pancreatic neuroendocrine tumor, functional, non-functional

## Abstract

**Simple Summary:**

Pancreatic neuroendocrine tumors (PNETs) represent a small portion of pancreatic neoplasms and can manifest as either functional or non-functional. Managing these tumors presents a challenge due to their rarity and heterogenous tumor biology. This manuscript offers a comprehensive review of PNET types, classification, and management strategies, with a focus on the evolving landscape of pancreatic surgery. It aims to assist clinicians in facilitating the management of these increasingly common tumors and optimizing patient outcomes.

**Abstract:**

The incidence of pancreatic neuroendocrine tumors (PNETs) is on the rise primarily due to the increasing use of cross-sectional imaging. Most of these incidentally detected lesions are non-functional PNETs with a small proportion of lesions being hormone-secreting, functional neoplasms. With recent advances in surgical approaches and systemic therapies, the management of PNETs have undergone a paradigm shift towards a more individualized approach. In this manuscript, we review the histologic classification and diagnostic approaches to both functional and non-functional PNETs. Additionally, we detail multidisciplinary approaches and surgical considerations tailored to the tumor’s biology, location, and functionality based on recent evidence. We also discuss the complexities of metastatic disease, exploring liver-directed therapies and the evolving landscape of minimally invasive surgical techniques.

## 1. Introduction

Pancreatic neuroendocrine tumors [PNETs] arise from islet cells of the pancreas that are scattered throughout the organ. They are rare tumors, with an annual incidence of approximately 1.8 in females to 2.6 in males per 100,000 population, and they account for about 2% of all pancreatic neoplasms [1,2]. The majority of patients with PNETs are diagnosed in the fifth to sixth decades of life and are oftentimes found in the pancreatic tail and head as opposed to the body [1,3,4,5].

Due to the improved quality and increased utilization of cross-sectional imaging, the incidence of PNETs has been on a steady rise as they are detected incidentally as hyperenhancing solitary masses [1,6]. Many of these incidentally found PNETS that are increasingly diagnosed represent non-functional lesions [NF-PNETs], while functional PNETs [F-PNETs] make up the minority of diagnosed lesions. The natural history of PNETs is highly variable and ranges from benign solitary masses to locally advanced or aggressive metastatic disease. The most widely accepted histopathological classification is the World Health Organization (WHO) classification that predicates on the lesion’s Ki-67 proliferation index and mitotic index [7].

Given the rising incidence of incidentally found NF-PNETs, understanding the biologic behavior of these tumors becomes increasingly important when deciding whether to manage these lesions operatively or non-operatively. This comprehensive review explores the intricate landscape of functional and non-functional PNETs, emphasizing their histological classification and staging and the evolving paradigms in diagnosis and surgical management.

## 2. Histologic Classification

The most widely accepted histologic classification system of PNETs is the 2019 WHO classification. According to the most recent classification system, well-differentiated neuroendocrine neoplasms (NEN) can be categorized into three groups (NET G1, NET G2, and NET G3) based on the Ki-67 proliferation index and mitotic index per high power field (HPF) on immunohistochemistry analysis. NET G1 tumors are characterized by a mitotic index of <2/10 HPF and a Ki-67 of <3%, NET G2 tumors are characterized by a mitotic index of 2–20/10 HPF and Ki-67 of 3–20%, and NET G3 tumors are characterized by a mitotic index of >20/10 HPF and Ki-67 of >20% (Table 1).

The terms “well-differentiated” or “poorly differentiated” are based on their molecular profiles; mutations in MEN1, DAXX, and ATRX genes are found in well-differentiated NETs, whereas mutations in TP53 and RB1 are found in poorly differentiated NECs. Conversely, poorly differentiated NENs are grouped as neuroendocrine carcinomas (NECs) and are considered high-grade by default. NECs are further classified into small-cell or large-cell based upon the histopathology. The final entity includes a mixed category (MiNEN) with characteristics of both well-differentiated NETs and poorly differentiated NECs [8].

## 3. Radiographic Diagnostics

Imaging modalities play a crucial role in the diagnosis and management of PNETs, providing clinicians with valuable insights into tumor localization, characteristics, and potential metastases. These diagnostics aid in the precise staging of tumors, facilitate surgical planning, and contribute to the ongoing evolution of tailored treatment strategies.

**Transabdominal ultrasound (US)**—While the sensitivity and specificity of US in the detection of liver metastases is high, its sensitivity in detecting the primary pancreatic tumor is only about 39%. However, the sensitivity and specificity increase with the use of contrast enhanced ultrasonography (CEUS) [9]. Ultrasound’s widespread availability, cost-effectiveness, and absence of ionizing radiation make it an appealing imaging modality. That said, its utility in clinical decision-making is constrained by the aforementioned lower sensitivity and its reliance on operator proficiency.

**Multidetector computed tomography (CT) scan**—A contrast-enhanced, pancreas protocol CT scan is one of the most informative diagnostic tools in the diagnosis of a PNET. PNETs are typically hyperenhancing on pancreatic protocol CT scans compared to pancreatic adenocarcinomas, which are typically hypoenhancing. The sensitivity and specificity of CT scan is about 73% and 96%, respectively [9]. Importantly, CT imaging provides superior granularity in the assessment of the location of the tumor and its relationship with adjacent visceral vessels when compared to other modalities (Figure 1a).

**Magnetic resonance imaging** (**MRI**) **with magnetic resonance cholangiopancreatography (MRCP)**—MRI with MRCP has been found to have similar sensitivities and specificities as CT scans, with the additional advantage of it being free of ionizing radiation and its tolerability in patients allergic to iodinated contrast. The other advantage of MRI with MRCP is the ability to better delineate the ductal anatomy around the pancreatic mass, which could aid in surgical planning when enucleation is being considered [9,10]. This often comes at the cost of decreased granularity in the assessment of major visceral vessel involvement. The classic appearance of PNETs on MRI is that of a peripherally hyperenhancing mass on the TI phase (Figure 1b).

**Endoscopic ultrasound (EUS)**—Among all diagnostic modalities, EUS has the highest sensitivity (77–100%) and specificity (98%) in the detection of pancreatic masses [9]. It is an important modality especially in the diagnosis of smaller F-PNETs in the setting of biochemical activity but the absence of any discernible pancreatic lesions on cross-sectional imaging as it better detects lesions smaller than 1 cm in size. Additionally, EUS can also provide an accurate assessment of the distance between the mass and pancreatic duct, which is critical for surgical planning when assessing for the appropriateness of enucleation. The other advantage of EUS is the ability to perform fine needle aspiration (FNA) and biopsy. The North American Neuroendocrine Tumor Society (NANETS) recommends EUS-guided FNA only in patients when the diagnosis of PNET is questionable or if there is a question regarding the tumor grade [11]. However, most institutions will perform FNA to determine tumor grade especially when the lesion in question is small.

**Radioisotope labelled imaging**—Various radioisotopes (indium, gallium, and technetium) have been used to identify pancreatic neuroendocrine tumors in conjunction with CT and positron emission tomography (PET) scanning. Gallium-labeled somatostatin analogs, when used with PET [^68^Ga DOTATATE PET/CT], have been found to have the highest sensitivity (81–100%) and specificity (90–100%) in diagnosing PNETs and metastatic disease [9,10,12]. The use of ^68^Ga-labeled somatostatin analogs is highly recommended as it could potentially upstage the disease and alter clinical management [9,13]. However, it is important to note that its utility in higher-grade and poorly differentiated PNETs are limited due to the associated loss in somatostatin receptors in these lesions. The use of ^18^F-FDG PET/CT, on the other hand, has been shown to increase the sensitivity of diagnosing these high-grade and poorly differentiated tumors and can complement somatostatin receptor scintigraphy [14,15].

**Intraoperative ultrasonography (IOUS)**—IOUS is an important intraoperative diagnostic adjunct that can be used to diagnose lesions in the pancreas and liver with a sensitivity of greater than 90% [9]. Intraoperative ultrasonography should always be used when parenchymal-sparing approaches are being considered and also in situations where there is a concern for multifocal disease such as in patients with familial syndromes (MEN-1).

## 4. Clinical Staging

The eighth edition of the American Joint Committee on Cancer (AJCC) staging is the most widely adopted and validated system currently used to stage PNETs according to tumor size (T), nodal status (N), and presence of distal metastasis (M) [16]. In the latest iteration of the staging manual, the TNM staging of PNETs does not apply to poorly differentiated NECs, which instead follows the staging of exocrine pancreatic adenocarcinoma. The invasion of adjacent structures including the stomach, spleen, colon, adrenal gland, and major visceral vessels such as the celiac axis or the superior mesenteric artery is now considered T4 disease.

While the tumor stage is based on AJCC classification as mentioned above, it is important to note that the main prognostic determinant for PNETs is the tumor grade. PNETs with low Ki-67 index confer a more favorable prognosis with improved recurrence-free and overall survival demonstrated in studies compared to PNETs with a high Ki-76 index [17,18,19,20]. The 5-year overall survival rates for NET G1, G2, and G3 are 100%, 95.5%, and 85.4%, respectively. In contrast, the 5-year overall survival rate for patients with NECs is 20%, which aligns more with the biologic behavior and prognosis of patients with pancreatic adenocarcinomas [21].

## 5. Non-Functional PNETs

NF-PNETs represent neuroendocrine tumors that do not produce hormones that lead to symptomatology. Up to 85% of all diagnosed PNETs are non-functional [2,22]. These lesions are most commonly sporadic but can also be a part of genetic syndromes such as Multiple Endocrine Neoplasia 1 (MEN1), Von Hippel-Lindau disease VHL, Neurofibromatosis1 (NF-1), and Tuberous Sclerosis (TSC) [23]. Mutations in genes regulating transcription and cell signaling pathways such as PI3K/AKT/mTOR and ATRX/DAXX leading to chromatin remodeling and telomere maintenance are involved in the formation and metastasis of NF-PNETs [23,24,25].

Most small NF-PNETs that are less than 2 cm are diagnosed incidentally on cross-sectional imaging [1,4,26]. NF-PNETs larger than 2 cm often present in a delayed fashion with mass-like compressive symptoms leading to abdominal pain, nausea, vomiting, or jaundice or in the form of metastatic disease [5,27,28]. Larger tumor size and higher grade are associated with higher metastatic occurrences, with the more common site of metastases being the liver [29,30]. Other sites of metastases include the lymph nodes, peritoneum, and bones, among others [30].

### 5.1. Diagnosis of Non-Functional PNETs

Diagnosing NF-PNETs relies on various imaging modalities described above to identify and characterize these mostly asymptomatic and incidentally detected tumors. Adjuncts including EUS-guided FNA or core needle biopsies can be used to differentiate NF-PNETs from other pancreatic lesions.

### 5.2. Role of Tumor Markers

Even though NF-PNETs are non-functional by definition, they have been found to secrete Chromogranin A and pancreatic polypeptide that have previously been used as circulating tumor markers [31]. It is important to note that serum Chromogranin A has a low specificity in the diagnosis of PNETs, with elevated serum levels also found in patients with renal failure, liver failure, chronic atrophic gastritis, and acute coronary syndrome and in the setting of proton pump inhibitors or H2-antagonist use. However, its trend in serum levels has proven beneficial, with recent ENETS guidelines recommending its use for evaluating tumor response, detecting tumor progression, and recurrence [13,32].

### 5.3. Surgical Management of NF-PNETs

#### 5.3.1. Sporadic Asymptomatic Tumors of Less Than 1 cm

In general, the decision to resect NF-PNETs is based on tumor size rather than the grade of the tumor on biopsy. The correlation between resected tumor size and grade on final pathologic examination is more accurate when compared to preoperative biopsy grade [33]. The incidence of small sporadic NF-PNETs is on the rise due to more of these tumors now being incidentally detected on cross-sectional imaging done for other reasons. Most NF-PNETs less than 1 cm in size have an indolent biologic behavior, with studies consistently showing the safety of “watchful waiting” for these smaller tumors without any difference in short- and long-term survival outcomes when compared to patients who underwent surgical resection [34,35,36,37]. Congruently, the NANETS, ENETS (European Neuroendocrine Tumor Society), and NCCN (National Comprehensive Cancer Society) guidelines unanimously recommend that sporadic NF-PNETs of <1 cm in size can be safely observed with active surveillance. There is, however, not a clear imaging protocol that can be followed for an active surveillance strategy. The NCCN guidelines recommend CT or MRI with contrast every 3 to 12 months for surveillance [38]. An ongoing multicenter prospective study from Denmark is evaluating a protocol that includes multimodal imaging techniques at various intervals. The interim analysis, however, showed that participating centers deviated from the surveillance protocol early on in the study owing to personal preferences and beliefs [37]. In the absence of a unifying consensus, we suggest the following protocol for surveillance of small non-functional neuroendocrine tumors (Figure 2). Surgical resection should be reserved for patients with tumor growth on surveillance imaging, evidence of nodal metastasis, or significant anxiety with repeated imaging under a “watchful waiting” strategy.

#### 5.3.2. Sporadic Asymptomatic Tumors of 1–2 cm

The clinical management for this subset of patients remains highly controversial. While contemporary studies have demonstrated survival superiority with surgical resection, patient selection bias remains with most of these studies representing retrospective analyses [35,36]. While ENETS recommends observation for 1–2 cm tumors, NANETS and NCCN guidelines did not reach a clear consensus and recommended an individualized approach [11,13,38]. The Canadian expert group recommends that only patients with a low Ki-67 index and without any vascular invasion or metastatic disease should be considered for surveillance [39]. In reality, the decision to resect or actively surveil patients with 1–2 cm PNETs should hinge on the patient’s life expectancy, competing medical risks such as comorbidities, tumor location that dictates extent of resection (pancreatoduodenectomy versus distal pancreatectomy), and patient preferences (Figure 2).

#### 5.3.3. Sporadic Asymptomatic Tumors of 2 cm or Larger

There is a notable correlation between tumor size and grade, with larger tumors being associated with more aggressive biologic behavior [21,33]. Larger tumor size is also associated with a 2.6-times higher risk of lymph node metastases that, in turn, is associated with poorer disease-free survival (5-year survival of 71.8% vs. 95.6% for tumors < 2 cm). Consequently, surgical resection is typically recommended for tumors exceeding 2 cm [11,13,38,40]. Historically, formal segmental resection in the form of pancreatoduodenectomy or distal pancreatectomy is performed to treat these lesions. However, parenchymal-sparing approaches such as enucleation and central pancreatectomy are gaining traction. They provide the benefit of preserving the exocrine and endocrine functions of the pancreas at the expense of higher short-term morbidity rates including a higher pancreatic fistula rate [41]. One concern in parenchymal-sparing approaches is the limited ability for lymph node sampling and, thus, the inability to assess for nodal metastases [11]. However, a recent study of four high-volume centers comparing parenchymal-sparing approaches with formal segmental resections for NF-PNETs < 3 cm in size demonstrated similar disease-free and overall survival. However, parenchymal-sparing approaches have the advantage of decreased blood loss, shorter operative times, and lower complication rates, solidifying parenchymal-sparing strategies as a safe approach for select tumors (Figure 2) [42].

Larger PNETs, especially when located in the pancreas head or uncinate, can also present as locally advanced disease when major visceral vessels are involved. There are typically two phenotypes: tumors that are well encapsulated that typically have a mass-like effect that “pushes” visceral vessels (Figure 3) and tumors that are more irregular and infiltrative, which tend to tether onto visceral vessels (Figure 4). The former phenotype can usually be separated from visceral vessels without the need for vascular resection and reconstruction, whereas the latter will typically require it. Nevertheless, vascular resection and reconstruction should be performed in appropriately selected patients given the typically favorable biology of PNETs.

#### 5.3.4. Tumors with Evidence of Nodal Metastasis

Nodal disease is significantly associated with tumor grade and size and found to be present in approximately 3% of grade 1, 16% of grade 2, and up to 100% of grade 3 tumors [43,44]. Ga-DOTATATE PET and EUS-guided FNA biopsy are considered the diagnostic tests of choice to assess nodal disease of PNETs preoperatively. While the presence of nodal metastasis is historically associated with poor overall survival, recent evidence demonstrates that parenchymal-sparing approaches to resection are not associated with inferior survival. This suggests that lymphadenectomy, even in the setting of node-positive disease, does not alter the oncologic outcomes for PNETs [44,45].

Congruently, studies have shown that extended regional lymphadenectomy has not been shown to improve overall survival, and no consensus was reached among reviewers at the AHPBA Delphi consensus process due to the increased risk associated with extended lymphadenectomies [46]. That said, both NCCN and ENETS still recommend formal oncological resections with lymphadenectomy for suspected node-positive disease (Figure 5) [13,38].

Patients experiencing local recurrence following surgical resection may undergo re-resection, although current data on outcomes for such circumstances are lacking. Management decisions for these patients depend on factors such as the extent of the disease, stability of disease progression, functional status, expected life expectancy, and surgeon’s expertise.

#### 5.3.5. Tumors with Liver Metastasis

PNETs most commonly metastasize to the liver, bone, and lung. Liver metastases have been associated with worse overall survival [47]. However, curative resection can be pursued for isolated metastasis with some improvement in overall survival if the primary lesion can be resected as well [48]. The pattern of liver metastasis is usually multiple small tumors involving both hepatic lobes (Figure 6) [49]. The NANETS guidelines did not reach a consensus on the utility of cytoreduction in this setting, and the evidence supporting resection remains scarce. However, cytoreduction should be considered if patients are symptomatic and debulking is able to improve their quality of life [11]. Synchronous surgical resection of hepatic metastases and primary tumor can be performed with acceptable low mortality in selected patients [49]. Single-staged pancreaticoduodenectomy and major hepatectomy is discouraged due to its associated morbidity and mortality rates [13]. Isolated liver metastasis from neuroendocrine tumors is currently an acceptable indication for liver transplantation. The Milan or UCSF criteria should be utilized to select these patients appropriately [50]. The 5-year overall survival ranges from 36 to 90%; however, the data is not uniform and is prone to selection bias due to the retrospective nature of published studies [51,52,53].

#### 5.3.6. Role of Liver-Directed Therapies for Liver Metastasis

Metastatic involvement of the liver significantly impacts patients’ quality of life, often leading to carcinoid syndrome, biliary obstruction, liver insufficiency, and abdominal pain (Figure 6) [54,55]. For patients experiencing symptoms but deemed not to be surgical candidates, various liver-directed therapies offer viable treatment options. These encompass percutaneous ablation techniques such as laser ablation (LA), radiofrequency ablation (RFA), microwave ablation (MWA), and irreversible electroporation (IRE). Additionally, endovascular treatments, including transarterial embolization (TAE), transarterial chemoembolization (TACE), and peptide receptor radionuclide therapy (PRRT), are also effective alternatives [51,56,57,58,59]. These modalities can be used alone or in combination with each other. In general, ablative techniques are effective and safe for oligometastatic disease. For diffuse disease with involvement of more than 75% of the liver, endovascular approaches and PRRT represent better alternatives, although they are limited by their availability at specialized centers [59].

**Figure 6 cancers-16-01501-f006:**
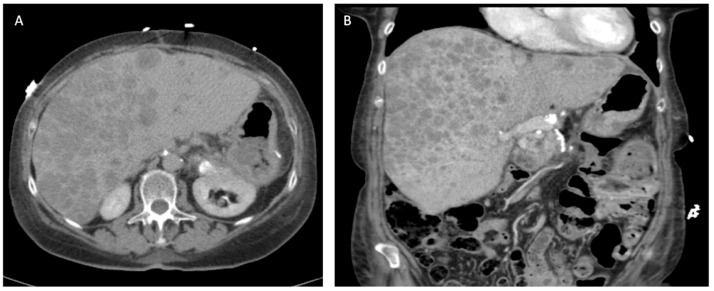
Unresectable liver metastases from pancreatic neuroendocrine tumor. (**A**) Axial view. (**B**) Coronal view.

#### 5.3.7. Open Versus Minimally Invasive Resection

The adoption of minimally invasive surgical (MIS) approaches for pancreatic surgery was initially slow secondary to concerns about oncologic adequacy, the associated steep learning curve, and the lack of specialized equipment. However, recent studies have addressed safety and oncologic outcomes with MIS approaches encompassing both laparoscopic and robotic platforms. A study by Zheng and colleagues encompassing over 1000 patients with PNETs demonstrated similar recurrence-free and overall survival in MIS pancreatectomies when compared to open procedures. Notably, the rate of clinically relevant postoperative pancreatic fistulas after pancreatoduodenectomies was lower in the MIS group (13% vs. 27%) [60]. Another study by Kim and colleagues focusing on NF-PNETs found that MIS approaches yielded comparable disease-free and overall survival to open approaches, with shorter hospital stays despite longer operative duration in the MIS group. Furthermore, lymph node yield was similar between the approaches [61]. Sutton and colleagues recently corroborated these findings in a multi-institutional registry involving 282 patients [62]. In conclusion, while high-quality randomized data is still lacking, the existing body of evidence supports the application of MIS approaches when feasible given their demonstrated superiority in postoperative outcomes as well as their equivalence in long-term oncologic outcomes.

#### 5.3.8. Role of Neoadjuvant and Adjuvant Therapies for High Grade PNETs/NECs

Neoadjuvant therapies, including both radiation and chemotherapy alone or in combination, have been utilized in several studies to downstage tumors, with the aim of improving R0 surgical margin rates and controlling disease progression when surgery is not feasible [63,64]. PRRT, [(177)Lu-DOTA(0),Tyr(3)]octreotate ((177)Lu-octreotate), and Capecitabine with Temozolomide (CAPTEM) are the most commonly used therapies in both neoadjuvant and adjuvant settings, showing varying degrees of favorable responses [64,65,66,67]. CAPTEM is currently the most widely used regimen, demonstrating efficacy in both progression-free and overall survival in a recent systematic review and meta-analysis [68].

Data on adjuvant therapy for poorly differentiated neuroendocrine tumors and NECs is limited and not well established. A variety of chemotherapy and immunotherapy options have become available but with limited efficacy. The current first-line therapy consists of platinum-based agents such as cisplatin or carboplatin with etoposide/irinotecan. Other options include FOLFOX, FOLFIRI, and CAPTEM [69]. Additionally, the SWOG S1609 trial explored the addition of immunotherapy with dual Anti-CTLA-4 and Anti-PD-1 blockade, demonstrating a 6-month progression-free survival rate of 32% and an overall response rate of 26% [70].

## 6. Functional PNETs

### 6.1. Insulinomas

Among F-PNETs, insulinomas are the most common, constituting up to 40% of all functional tumors [71,72]. According to the Mayo Clinic series, the estimated incidence of insulinomas is about 0.4 per 100,000 person-years [73]. Insulinomas are clinically suspected based on recurrent neuroglycopenic symptoms and hypoglycemia. The majority of insulinomas present as single benign tumors, with less than 10% presenting as multiple or malignant tumors [74]. Malignant insulinomas are typically aggressive and are associated with a poor prognosis (5-year survival of 55.6% vs. 10-year survival of 88% for benign insulinomas) [73,75,76].

#### 6.1.1. Diagnosis

The original account of hypoglycemia with hyperinsulinemia and symptoms was given by the father of pancreatic surgery, Dr. Allen O. Whipple, in 1935 [77]. Whipple’s triad includes the presence of neuroglycopenic symptoms (palpitations, diaphoresis, sweating, confusion, visual disturbances), hypoglycemia (plasma glucose < 50 mg/dL), and resolution of symptoms on the administration of glucose either orally or intravenously [74]. The sensitivity and specificity of Whipple’s triad alone to diagnose hypoglycemia secondary to insulinoma are not known, and it has limitations in asymptomatic hypoglycemia or patients with neurological damage [78].

Recreating hypoglycemic symptoms through a 72-h fast after a mixed meal has become the gold standard for diagnosing insulinomas. Clinical signs of hypoglycemia with plasma glucose concentrations of 55 mg/dL or lower, insulin levels of 3 μU/mL or higher, C-peptide levels of at least 0.6 ng/mL, and proinsulin levels of at least 5.0 pmol/L, along with decreased β-hydroxybutyrate levels of 2.7 mmol/L or less, suggest high endogenous insulin production. Evidence of high endogenous insulin production and an increase in plasma glucose of 25 mg/dL or more after the administration of intravenous glucagon indicate the presence of an insulinoma [13,79]. Up to 95% of patients will develop symptoms within the first 48 h, and a full 72-h fast is typically not necessary [80].

Efforts to localize an insulinoma typically include a pancreas protocol CT scan or MRI. On CT or MRI, insulinomas usually appear homogeneous with intense enhancement during the arterial phase. Up to 90% of insulinomas can be detected on the CT scan [10,81]. However, for the minority of insulinomas that are not appreciated on cross-sectional imaging, more invasive approaches such as EUS and arterial stimulation venous sampling (ASVS) with angiography may be considered when there is a high clinical suspicion of the diagnosis in the setting of negative imaging. EUS is highly sensitive and specific for insulinomas with characteristic features, including homogeneously hypoechoic, round shape, and well-defined margins. An additional advantage of EUS is the ability to perform EUS-guided FNA of the primary tumor and any suspicious peripancreatic lymph nodes [74].

IOUS is also an important adjunct when attempting to localize and resect these smaller lesions in the operating room. ASVS, on the other hand, is the most accurate test for localizing insulinomas with a pooled sensitivity of 93% and specificity of 86% [82,83]. However, with the improvement in less invasive approaches such as cross-sectional imaging and its improved ability to detect insulinomas, ASVS has fallen out of favor and is only reserved as a last-resort diagnostic maneuver. On ASVS combined with angiography, insulinomas typically present as well-defined, hypervascular round or oval blushes [83,84]. Typically, a 10% calcium gluconate injection at a dose of 0.01–0.025 meq/kg of calcium diluted with normal saline is injected into the selectively catheterized artery (splenic artery, superior mesenteric artery, or gastroduodenal artery), and blood samples are obtained from the hepatic vein at different time intervals. An increase of twofold or more of the insulin levels from the baseline is considered diagnostic. Arterial catheterization of the gastroduodenal artery, superior mesenteric artery, and splenic artery can be performed to detect insulinomas in the pancreas head, body, and tail, respectively [84].

#### 6.1.2. Management

Surgery is recommended for all insulinomas given their symptom profile. After localization, parenchymal-sparing procedures such as enucleation or central pancreatectomy can be considered when appropriate and safely performed with excellent surgical outcomes and a low recurrence rate of about 3% [74,85,86]. When compared to formal resections, enucleations have a higher reoperation rate and clinically significant postoperative pancreatic fistula rates. Minimally invasive resections are now being performed more frequently at high-volume centers with comparable outcomes [86]. Irrespective of the approach, negative margins are the goal of surgical resection. In patients with lesions larger than 4 cm or close to the main pancreatic duct, segmental resection should instead be considered [75].

In patients with insulinomas that are suboptimal surgical candidates, endoscopic ultrasound-guided radiofrequency ablation (EUS-RFA) may be considered. While the exact role of EUS-RFA in the management of these lesions needs to be better studied, the literature suggests that it is an effective treatment option in smaller (<18 mm) insulinomas [87,88]. In a study conducted by Crinò and colleagues, endoscopic ultrasound (EUS) demonstrated a lower rate of adverse events and shorter hospital stays compared to surgery while maintaining similar clinical efficacy. The mean size of insulinomas in this study was 13.5 mm [89]. A multicenter randomized controlled trial comparing EUS with surgery for insulinomas smaller than 2 cm is currently ongoing, promising further insights into the efficacy of EUS [90]. In the setting of multifocality or metastatic spread of insulinomas, familial syndromes such as MEN-1 need to be considered [91]. Surgical resection of multiple MEN-1associated insulinomas has been shown to be safe and effective, yielding a higher 10-year symptom-free survival rate compared to medically managed patients [91]. However, given the complex nature of metastatic insulinomas and the paucity of large-scale studies, a multidisciplinary approach involving surgical, medical, and interventional strategies should be considered and the treatment plan individualized based on the patient’s clinical presentation, tumor characteristics, and overall health status.

### 6.2. Gastrinomas

Gastrinomas are the second most common F-PNET with a reported incidence ranging from 0.5 to 2 cases per million population [92]. While a majority of gastrinomas are sporadic, up to 30% are associated with MEN-1. In fact, gastrinomas are the most common F-PNET associated with MEN-1 [93,94].

#### 6.2.1. Diagnosis

A diagnosis of gastrinoma requires an elevated fasting gastrin level over 10-fold above normal (>100 pg/mL) and a gastric pH of less than 2 [95,96]. Patients should discontinue proton pump inhibitors for 1 week and H2-blockers for 48 h before testing to avoid confounding gastric acid levels [94,97]. Additionally, disproportionately high gastrin levels are also correlated with tumor size and the presence of metastases, and their measurement is crucial for diagnostic insight [95]. However, up to 40% of patients will have fasting serum gastrin levels that are less than 10-fold of the upper limit. This population, along with the rising use of proton pump inhibitors, poses a diagnostic challenge. In these cases, a secretin stimulation test can be used to differentiate Zollinger–Ellison syndrome (ZES) from other probable causes of hypergastrinemia [96]. Secretin stimulation test have been reported to have a sensitivity of 94% and specificity of up to 100% [95]. For a secretin stimulation test, fasting serum gastrin levels are measured first. Intravenous synthetic secretin is then administered, and blood samples are withdrawn at 2, 5, 10, and 15 min for the measurement of gastrin levels. Secretin usually inhibits the formation of gastrin from the G-cells. However, in patients with gastrinomas, the gastrin levels paradoxically continue to rise or remain elevated after secretin administration, with levels more than 120 pg/mL being diagnostic of ZES [94].

After a biochemical diagnosis of ZES has been made, localization of gastrinoma is similar to that of insulinomas. Non-invasive imaging methods including CT or MRI, somatostatin receptor scintigraphy, and ^68^Ga DOTATATE PET/CT can be used for localization. ^68^Ga DOTATATE PET/CT has the advantage of detection of potential distant metastases, including liver, lung, and bone lesions with high sensitivity and specificity of up to 100% [94,98]. When localization efforts come up short in the setting of strong clinical suspicion, invasive approaches including EUS and selective arterial secretin injection (SASI) can be considered. The advantages and limitations of EUS have been previously explained in the review. SASI is an angiographic test similar to ASVS used for insulinomas in which secretin is injected into the feeding vessels supplying the gastrinoma and gastrin levels are measured in the hepatic vein at 20, 40, 60, 90, and 120 s post injection [99]. An increase of 80 pg/mL over 40 s is used as criteria for diagnosis of metabolically active gastrinoma [94].

#### 6.2.2. Management

The general consensus from ENETs and NANETs guidelines is that all patients with sporadic gastrinomas should be treated with surgical resection whenever feasible as surgical resection has been found to be associated with a more favorable overall survival and a cure rate of up to 60% [11,13,100]. Unlike insulinomas, approximately 98% of gastrinomas can be detected during surgical exploration [100]. Vascular resection and reconstruction can be performed at experienced centers safely if necessary [101]. As a proportion of patients with gastrinomas are malignant and/or have nodal involvement, all guidelines recommend adequate assessment of locoregional lymph nodes. For the same reasons, a parenchymal-sparing approach is not recommended given its inability to adequately assess peripancreatic lymph nodes [98].

Patients with ZES-associated with MEN-1 often present with smaller tumors but multifocal disease. Surgical treatment of MEN-1 associated gastrinomas is usually not recommended because of the generally higher recurrence rate given disease multifocality [13,102,103]. Surgical resection is typically only reserved for when the tumor size is more than 2 cm or if there is rapid progression over a period of 6 to 12 months [13,94]. It is important to note that parathyroidectomy is usually recommended first in patients with MEN-1 who are being considered for surgery as hypercalcemia could lead to increased gastrin production [94].

Unfortunately, up to one-third of the patients present with liver metastasis at the time of diagnosis. Cytoreductive debulking procedures should only be performed if up to 70% of the disease can be safely resected, as that has been shown to improve overall symptoms and 5-year survival [11,104]. Liver-directed therapies including RFA, TAE, TACE, or PRRT that have been previously described can be considered whenever feasible for symptom palliation. Additionally, gastrinomas have been found to express high SSTR expression, making them susceptible to somatostatin analog therapy [105]. The use of somatostatin analogs in unresectable gastrinomas is based on data extrapolated from the CLARINET and PROMID trials showing benefit in terms of progression-free survival in patients with metastatic neuroendocrine tumors [106,107]. The role of chemotherapy for metastatic gastrinomas is currently not clear due to the paucity of gastrinoma-specific studies.

### 6.3. Glucagonoma

Glucagonoma is a rare PNET originating from the alpha cells of the pancreatic islets that secrete glucagon. The true incidence of glucagonomas remains unknown, but it reportedly constitutes up to 7% of all pancreatic neuroendocrine tumors [108]. The glucagonoma syndrome associated with these tumors presents with distinctive features, including dermatitis (necrolytic migratory erythema) observed in approximately 90% of patients, along with weight loss, diabetes mellitus, diarrhea, and mucosal inflammation [108,109,110,111]. Additional laboratory findings include anemia and hypoaminoacidemia. Notably, most patients present with metastatic disease at the time of diagnosis [112]. While primarily sporadic, molecular studies reveal inactivating mutations of MEN-1, Rb, and p53 in a substantial number of cases [113,114,115].

#### 6.3.1. Diagnosis

The aforementioned clinical features of glucagonoma along with elevated fasting serum glucagon levels (>500 pg/dL) and the identification of a pancreatic mass essentially clinches the diagnosis of glucagonoma. Supporting findings, such as elevated serum glucose levels, increased serum insulin and C-peptide, and elevated hemoglobin A1C indicating glucose intolerance are other clinical manifestations [116]. Localization of glucagonoma follows the principles described earlier for insulinomas and gastrinomas. Localization strategies can include non-invasive imaging (CT, MRI, somatostatin receptor scintigraphy, ^68^Ga DOTATATE PET/CT). If initial imaging is inconclusive, EUS/FNA and selective visceral angiography may be employed to locate the pancreatic tumor with more granularity.

#### 6.3.2. Management

The management of glucagonomas requires a comprehensive approach encompassing symptom control, surgical intervention, and metastatic disease management if tumor spread is present. Symptom control often involves the use of somatostatin analogs (octreotide, lanreotide) due to the high prevalence of SSTRs on glucagonomas [106,107,117]. Surgical exploration, whenever feasible, is recommended even when no tumor is found on initial imaging due to high cure rates; Norton and colleagues found that almost all tumors not readily apparent on preoperative imaging could eventually be found on surgical exploration [100,101]. Similar to patients with gastrinomas, parenchymal-sparing procedures are not recommended due to the high malignant potential and presence of silent metastasis at the time of diagnosis [11].

In the setting of metastatic disease, the treatment goal centers on symptom control with medical management-targeted ablative therapy. This can include somatostatin analogues, hepatic artery embolization, and PRRT. Chemotherapeutic regimens, guided by molecular profiling to identify targets such as tyrosine kinase and mTOR, have demonstrated variable responses. Notably, improved progression-free survival has been observed with the use of drugs targeting these molecular pathways, either alone or in combination with somatostatin analogues [118].

### 6.4. VIPoma

VIPomas, short for vasoactive intestinal peptide secreting neuroendocrine tumors, are rare PNETs associated with classic findings of watery diarrhea, hypokalemia, and achlorhydria (WDHA syndrome). This was first reported in 1958 and is also known as Verner–Morrison syndrome [119]. The incidence of VIPoma is estimated to be about 1 per million population [120]. Vasoactive intestinal peptide stimulates the exocrine pancreas and small bowel smooth muscles, as well as inhibits secretion of acid in the stomach [121,122]. Severe refractory watery diarrhea is the hallmark of a VIPoma, with other less characteristic findings of facial flushing, secondary hypokalemia, and achlorhydria [123,124]. VIPomas are usually solitary sporadic tumors and are most commonly found in the tail of the pancreas [123,124,125]. Up to 10% of VIPomas are extrapancreatic [125,126]. Metastases, frequently to the liver, is prevalent in up to 50% of patients [123,124,125]. With such a high proportion of patients presenting with late-stage disease on presentation, prognosis is typically poor with studies reporting a median disease-free survival of just 16 months [124].

#### 6.4.1. Diagnosis

Normal plasma levels of VIP are below 70 pg/mL. To establish a VIPoma diagnosis, high-volume secretory diarrhea with a low osmotic gap (<50 mOsm/kg), along with characteristic laboratory findings of hypokalemia and achlorhydria, is needed. Elevated levels of VIP (>75 pg/mL) would further confirm the diagnosis [126,127]. A majority of VIPomas are more than 2 cm in diameter and are easily visible on cross-sectional abdominal imaging [123,124,128]. Previously mentioned non-invasive and invasive testing including CT, MRI, somatostatin receptor scintigraphy, ^68^Ga DOTATATE PET/CT, and EUS/FNA can also be used to localize a VIPoma. Metastatic workup including CT of chest, abdomen, and pelvis should be completed to rule out metastatic disease given the typical late-stage disease on presentation.

#### 6.4.2. Management

The treatment of VIPomas typically begins with management of the presenting symptoms, including intravenous fluid resuscitation, electrolyte replacements, and anti-diarrheal agents. Somatostatin analogs should be started as soon as the diagnosis is suspected as they can help with early symptom control [129,130]. Once symptom control has been achieved and a pancreatic lesion has been localized in the absence of metastases, surgical resection should be considered expeditiously [131]. Segmental, oncologic resection with lymphadenectomy should be performed; parenchymal-sparing approaches are not recommended [11,123,124,128]. As most of the tumors are found in the tail of the pancreas, distal pancreatectomy with or without a splenectomy is the most common surgical procedure performed [124]. Somatostatin analogs should be utilized perioperatively for the prevention of cardiovascular complications [132].

Just like glucagonomas, the majority of VIPomas will present with advanced disease, and a multimodal treatment strategy should similarly be used for optimal management, including somatostatin analogs, chemotherapy (streptozotocin and 5-FU), targeted therapies (everolimus and sunitinib), and locoregional ablation of metastatic disease [106,107,127]. Data on the efficacy of surgical resection of metastatic disease is limited; however, some studies have shown benefit in overall survival when a complete resection was feasible [125].

### 6.5. Somatostatinoma

Somatostatin is a hormone secreted by delta (δ) cells of the pancreas and plays a role in modulating beta cell activity for glucose homeostasis [133]. Somatostatinomas are exceptionally rare PNETs, occurring in approximately 1 in 40 million individuals [134]. They are typically located in the pancreas or the peri-pancreatic duodenum. Notably, a significant proportion (~71%) is found at the ampulla of Vater, as reported by Garbrecht and colleagues [135].

Somatostatinoma syndrome is characterized by elevated somatostatin levels, diabetes, gallstones, achlorhydria, anemia, weight loss, and steatorrhoea, although this is a rare occurrence [134,135,136]. Mass effects causing obstructive jaundice or acute pancreatitis may occur when the tumor obstructs the ampulla [135]. Tumors can be sporadic or associated with neurofibromatosis type 1 (NF-1) mutation or Pacak–Zhuang syndrome (*EPAS1* gene encoding HIF) [136,137]. Like gastrinomas and VIPomas, up to 70% of somatostatinomas have metastatic disease at the time of diagnosis [13].

#### 6.5.1. Diagnosis

The diagnosis of somatostatinoma can be made when symptoms of somatostatinoma syndrome are present along with elevated somatostatinoma levels in the blood [136,138]. Other neuroendocrine hormones such as chromogranin A, neuro specific enolase, and 5- hydroxytryptamine may also aid in the diagnosis [135]. Various imaging modalities, such as CT, MRI, somatostatin receptor scintigraphy (Octreoscan), and 68Ga (Gallium)-DOTATATE, are similarly helpful for tumor localization.

#### 6.5.2. Management

Octreotide analogs are considered first-line therapies to help control symptoms and impede progression. Their utility is based on data extrapolated from CLARINET and PROMID studies [106,107]. Surgical resection remains the opportunity for cure for patients with somatostatinomas and has been shown to be associated with improvement in overall survival [135]. While parenchymal-sparing approaches are generally not recommended, minimally invasive techniques may be considered, though data on efficacy is limited [108,109].

Liver-directed therapies that have been previously mentioned can also be used for somatostatinomas with metastases to the liver [135]. Unfortunately, poorly differentiated tumors with extensive metastases exhibit poor overall survival despite surgical resection or chemotherapy [135].

## 7. Conclusions

PNETs represent a unique and challenging spectrum of diseases, encompassing a variety of histological classifications, clinical presentations, and treatment approaches. Surgical management strategies, tailored to tumor size and characteristics, have undergone paradigm shifts. The growing trend towards watchful waiting for small NF-PNETs emphasizes the importance of an individualized approach. Similarly, the decision to perform parenchymal-sparing pancreatectomy should be made with an understanding of specific NET subtype biology and tumor characteristics. In the context of metastatic disease, liver-directed and -targeted therapies contribute to the expanding tools available to clinicians. A patient-centered, multidisciplinary approach remains essential in the management of PNETs.

## Figures and Tables

**Figure 1 cancers-16-01501-f001:**
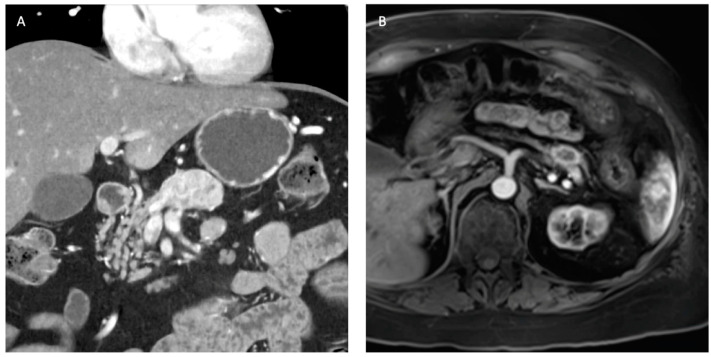
Classic radiographic appearances of pancreatic neuroendocrine tumors on cross-sectional imaging. (**A**) Hyperenhancing mass on arterial phase of computed tomography. (**B**) Peripherally hyperenhancing mass on T1 phase of magnetic resonance imaging.

**Figure 2 cancers-16-01501-f002:**
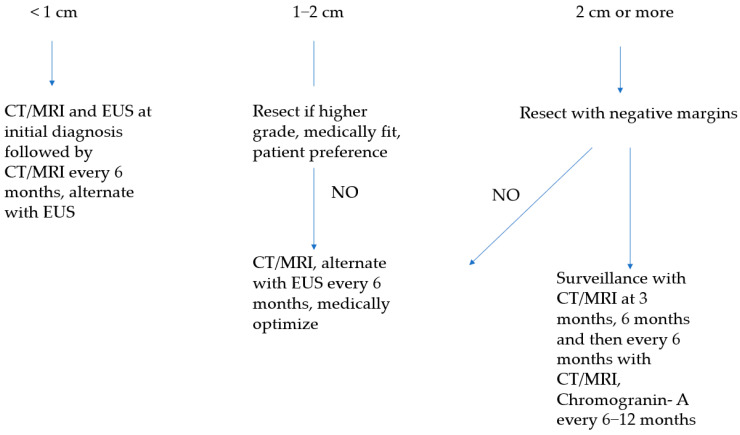
Proposed management algorithm for sporadic, non-functional pancreatic neuroendocrine tumors without metastatic disease.

**Figure 3 cancers-16-01501-f003:**
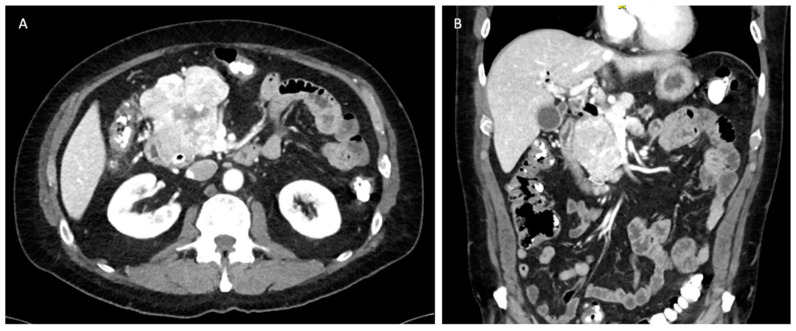
Radiographic appearance of borderline resectable pancreatic neuroendocrine tumors that has a more mass-like impact on adjacent visceral vessels. The patient did not require a venous resection or reconstruction. (**A**) Axial view. (**B**) Coronal view.

**Figure 4 cancers-16-01501-f004:**
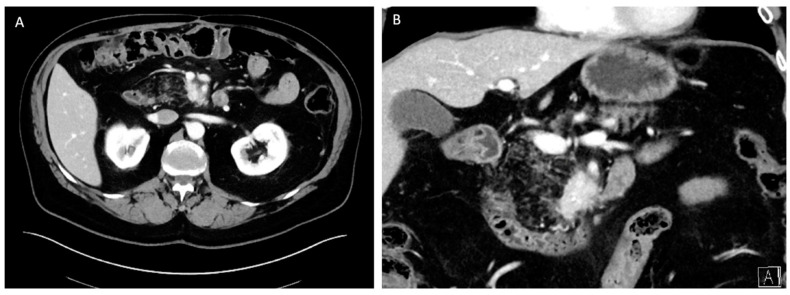
Radiographic appearance of borderline resectable pancreatic neuroendocrine tumors that has more infiltrative impact on adjacent visceral vessels. The patient required a superior mesenteric vein resection and reconstruction for margin-negative extirpation. (**A**) Axial view. (**B**) Coronal view.

**Figure 5 cancers-16-01501-f005:**
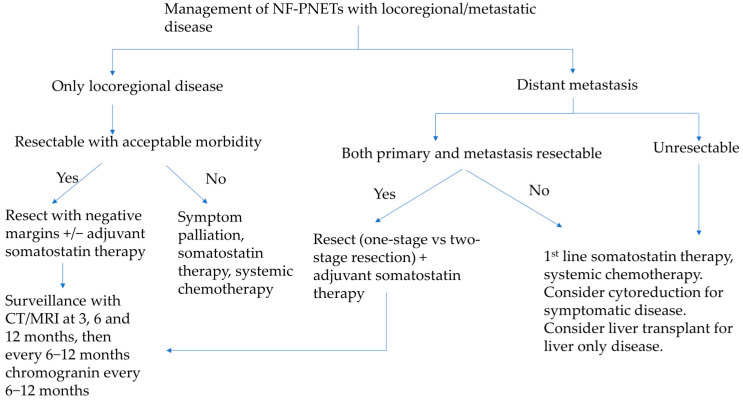
Proposed management algorithm for sporadic, non-functional pancreatic neuroendocrine tumors with locoregional or metastatic disease.

**Table 1 cancers-16-01501-t001:** The 2019 WHO histologic classification for pancreatic neuroendocrine tumor. Adapted from [8].

Terminology	Differentiation	Grade	Mitotic Rate * (Mitoses/2 mm^2^)	Ki-67 Index *
NET, G1	Well differentiated	Low	<2	<3%
NET, G2		Intermediate	2–20	3–20%
NET, G3		High	>20	>20%
NEC, small-cell type (SCNEC)	Poorly differentiated	High ^†^	>20	>20%
NEC, large-cell type (LCNEC)			>20	>20%
MiNEN	Well or poorly differentiated ^‡^	Variable ^‡^	Variable ^‡^	Variable ^‡^

LCNEC, large-cell neuroendocrine carcinoma; MiNEN, mixed neuroendocrine neoplasm; NEC, neuroendocrine carcinoma; NET, neuroendocrine tumor; SCNEC, small-cell neuroendocrine carcinoma. * Mitotic rates are to be expressed as the number of mitoses/2 mm^2^ as determined by counting in 50 fields of 0.2 mm^2^ (i.e., in a total area of 10 mm^2^); the Ki-67 proliferation index value is determined by counting at least 500 cells in the regions of highest labeling (hot-spots), which are identified at scanning magnification; the final grade is based on whichever of the two proliferation indexes places the neoplasm in the higher-grade category. ^†^ Poorly differentiated NECs are not formally graded but are considered high-grade by definition. ^‡^ In most MiNENs, both the neuroendocrine and non-neuroendocrine components are poorly differentiated, and the neuroendocrine component has proliferation indices in the same range as other NECs, but this conceptual category allows for the possibility that one or both components may be well differentiated; when feasible, each component should therefore be graded separately.

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
