# Peer review of "Contemporary Approaches to the Surgical Management of Pancreatic Neuroendocrine Tumors"

_cancers, 2024, doi:10.3390/cancers16081501_

Round 1

Reviewer 1 Report

Comments and Suggestions for Authors

Nicely written review on a cutting-edge topic. The authors should add some comments on the latest advancements on the use of non-surgical ablative treatments for pNETS, particularly EUS-RFA. On this regard, the authors should comment the results of a recent paper (PMID: 36871765) and should mention that a RCT is currently ongoing on this comparison

Author Response

Thank you for taking the time to review our manuscript titled "Contemporary Approaches to the Surgical Management of Pancreatic Neuroendocrine Tumors". We appreciate your valuable comments and suggestions, and we have addressed them as follows:

Reviewer #1 requested that we comment on the latest advancement on the use of non-surgical ablative treatments for PNETs particularly EUS-RFA.  We have added that in the insulinoma section as EUS-RFA has mainly been studied for the treatment of insulinomas. We have also commented and cited the recommended paper titled “Endoscopic ultrasound-guided radiofrequency ablation versus surgical resection for treatment of pancreatic insulinoma”. Comment on the ongoing prospective trial has also been added to the body of the manuscript as below.

“In a study conducted by Crinò and colleagues endoscopic ultrasound (EUS) demonstrated a lower rate of adverse events and shorter hospital stays compared to surgery, while maintaining similar clinical efficacy. The mean size of insulinomas in this study was 13.5 mm.[89] A multicenter randomized controlled trial comparing EUS with surgery for insulinomas smaller than 2 cm is currently ongoing, promising further insights into the efficacy of EUS.[90]”

Lastly, the following additions/changes have been done – inclusion of summary paragraph, changes in reference style, addition of author contributions, statement of funding, and correction of minor mistakes. All these changes have been highlighted in the updated manuscript.

Once again, we appreciate your valuable feedback and suggestions, which have helped us to improve our manuscript. We hope that our revised manuscript addresses your concerns adequately. If you have any further questions or comments, please do not hesitate to contact us.

Reviewer 2 Report

Comments and Suggestions for Authors

This is a very fine review on surgery for pancreatic NET

Nearly all aspects are addressed.

Some items should be addressed additionally:

-the role of neoadjuvant therapy for NET G3 or NEC, that are borderline resectable or locally advanced

- the role of surgery in local recurrence aside the liver. For example retroperitoneal, paraortal LN

minor: in double written in line 66

Author Response

Thank you for taking the time to review our manuscript titled "Contemporary Approaches to the Surgical Management of Pancreatic Neuroendocrine Tumors". We appreciate your valuable comments and suggestions, and we have addressed them as follows:

Reviewers #2&3 asked to address the role of neoadjuvant therapy for NET G3 or NEC. We added these sections in the manuscript describing the role of neoadjuvant chemo-radiotherapy, PRRT, and chemotherapy agents such as CAPTEM.

Role of Neoadjuvant and Adjuvant therapies for high grade PNETs / NECs

Neoadjuvant therapies, including both radiation and chemotherapy alone or in combination, have been utilized in several studies to downstage tumors, with the aim of improving R0 surgical margin rates and to control disease progression when surgery is not feasible.[63,64] PRRT, [(177)Lu-DOTA(0),Tyr(3)]octreotate ((177)Lu-octreotate), and Capecitabine with Temozolomide (CAPTEM) are the most commonly used therapies in both neoadjuvant and adjuvant settings, showing varying degrees of favorable responses.[64–67] CAPTEM is currently the most widely used regimen, demonstrating efficacy in both progression-free and overall survival in a recent systematic review and meta-analysis.[68]

Data on adjuvant therapy for poorly differentiated neuroendocrine tumors and NECs is limited and not well established. A variety of chemotherapy and immunotherapy options have become available but with limited efficacy. The current first line therapy consists of platinum based agents such as cisplatin or carboplatin with etoposide/irinotecan. Other options include FOLFOX, FOLFIRI and CAPTEM.[69] Additionally, the SWOG S1609 trial explored the addition of immunotherapy with dual Anti-CTLA-4 and Anti-PD-1 blockade, demonstrating a 6-month progression-free survival rate of 32% and an overall response rate of 26%.[70]  

The reviewer#2 asked for the role of surgery in local recurrence in sites other than the liver. Unfortunately, we were not able to find any data in the literature that can provide an evidence-based response. This is likely due to lack of specific studies addressing this issue. The management of these patients is highly dependent on patient’s extent of disease, cancer biology over time, functional status, expected life-expectancy and surgeon expertise. We have added these comments to the manuscript.

Patients experiencing local recurrence following surgical resection may undergo re-resection, although current data on outcomes for such circumstances are lacking. Management decisions for these patients depends on factors such as the extent of the disease, stability of disease progression, functional status, expected life expectancy, and the surgeon's expertise

Reviewer#2 pointed out a small mistake in the table 1 legend which has been corrected.

Lastly, the following additions/changes have been done – inclusion of summary paragraph, changes in reference style, addition of author contributions, statement of funding, and correction of minor mistakes. All these changes have been highlighted in the updated manuscript.

Once again, we appreciate your valuable feedback and suggestions, which have helped us to improve our manuscript. We hope that our revised manuscript addresses your concerns adequately. If you have any further questions or comments, please do not hesitate to contact us.

Reviewer 3 Report

Comments and Suggestions for Authors

Title: Surgical management of Pancreatic Neuroendocrine Tumors

This paper describes functional and non-functional PNETs, emphasizing their histological classification, staging, and the evolving paradigms in diagnosis and surgical management.

This paper describes a relatively rare case, but there are some questions and the author is requested to add the description according to the comments as below.

Major points

1.     Therapies for NEC

Patients with NEC have limited treatment options due to their poor prognosis. The author should describe details of the treatment of NEC.

Author Response

Thank you for taking the time to review our manuscript titled "Contemporary Approaches to the Surgical Management of Pancreatic Neuroendocrine Tumors". We appreciate your valuable comments and suggestions, and we have addressed them as follows:

Reviewer #3 asked to address the role of neoadjuvant therapy for NET G3 or NEC. We added these sections in the manuscript describing the role of neoadjuvant chemo-radiotherapy, PRRT, and chemotherapy agents such as CAPTEM.

Role of Neoadjuvant and Adjuvant therapies for high grade PNETs / NECs

Neoadjuvant therapies, including both radiation and chemotherapy alone or in combination, have been utilized in several studies to downstage tumors, with the aim of improving R0 surgical margin rates and to control disease progression when surgery is not feasible.[63,64] PRRT, [(177)Lu-DOTA(0),Tyr(3)]octreotate ((177)Lu-octreotate), and Capecitabine with Temozolomide (CAPTEM) are the most commonly used therapies in both neoadjuvant and adjuvant settings, showing varying degrees of favorable responses.[64–67] CAPTEM is currently the most widely used regimen, demonstrating efficacy in both progression-free and overall survival in a recent systematic review and meta-analysis.[68]

Data on adjuvant therapy for poorly differentiated neuroendocrine tumors and NECs is limited and not well established. A variety of chemotherapy and immunotherapy options have become available but with limited efficacy. The current first line therapy consists of platinum based agents such as cisplatin or carboplatin with etoposide/irinotecan. Other options include FOLFOX, FOLFIRI and CAPTEM.[69] Additionally, the SWOG S1609 trial explored the addition of immunotherapy with dual Anti-CTLA-4 and Anti-PD-1 blockade, demonstrating a 6-month progression-free survival rate of 32% and an overall response rate of 26%.[70]  

Lastly, the following additions/changes have been done – inclusion of summary paragraph, changes in reference style, addition of author contributions, statement of funding, and correction of minor mistakes. All these changes have been highlighted in the updated manuscript.

Once again, we appreciate your valuable feedback and suggestions, which have helped us to improve our manuscript. We hope that our revised manuscript addresses your concerns adequately. If you have any further questions or comments, please do not hesitate to contact us.

Round 2

Reviewer 1 Report

Comments and Suggestions for Authors

The revised version of the paper is OK. Thank you!

Reviewer 3 Report

Comments and Suggestions for Authors

Title: Surgical management of Pancreatic Neuroendocrine Tumors

This paper describes functional and non-functional PNETs, emphasizing their histological classification, staging, and the evolving paradigms in diagnosis and surgical management.

This paper describes a relatively rare case, but there are some questions and the author is requested to add the description according to the comments as below.

The authors responded to all comments and improved the manuscript.